# Timber Losses during Harvesting in Managed *Shorea robusta* Forests of Nepal

Upendra Aryal [1], Prem Raj Neupane [2,3,*], Bhawana Rijal [4] and Michael Manthey [1]

1 Institute of Landscape Ecology and Nature Conservation, University of Greifswald, Soldmannstraße 15, Room 1.40, 17489 Greifswald, Germany; aryal.upendra37@gmail.com (U.A.); manthey@uni-greifswald.de (M.M.)
2 Institute of Wood Science, World Forestry, University of Hamburg, Leuschnerstr. 91e, 21031 Hamburg, Germany
3 Friends of Nature, P.O. Box 23491, Kathmandu 44600, Nepal
4 Division Forest Office, Palpa, Tansen 6, P.O. Box 32500, Palpa 32505, Nepal; rijalbhawana1@gmail.com
* Correspondence: prem.raj.neupane@uni-hamburg.de; Tel.: +49-176-636-532-17

**Abstract:** Logging and sawing of timber using conventional tools by unskilled workers causes enormous damage to the valuable timber, residual stand, regeneration, and forest soil in Nepal. The purpose of this study was to find out the volume reduction factor and identify major strategies to reduce timber losses in the tree harvesting process in the *Terai Shorea robusta* forest of Nepal. Field measurements and product flow analysis of 51 felled trees from felling coupes and randomly selected 167 sawed logs were examined to study harvesting losses. Responses from 116 forest experts were analyzed to explore strategies for reducing harvesting and processing losses. The results showed that timber losses in the felling and bucking stage with and without stem rot were 23% and 22%, respectively. Similarly, timber losses in the sawing stage with and without stem rot were 31% and 30%, respectively. Paired *t*-test at 5% level of significance revealed that there was significant loss in both tree felling and log sawing stages with present harvesting practice. The most leading factor contributing to timber loss in all of the three stages was the use of inappropriate equipment during tree harvesting. Use of synthetic ropes for directional felling and skidding as well as flexible and portable sawing machine with size adjustment options during sawing were mainly recommended as strategies to reduce timber losses. This study serves as a baseline study to identify and quantify timber losses in different stages of tree conversion and also formulate their reduction strategies in Nepal.

**Keywords:** harvesting tools; reduction strategies; Terai *Shorea robusta* forest; tree harvesting losses

## 1. Introduction

Rapid enhancements in long-term forest production can be achieved only with well-devised harvesting operations and stand improvement treatments through sustainable forest management in tropical forests [1]. Unplanned forest tree harvesting results in higher amount of timber losses in tree felling [2] and in sawing phases [3]. A high amount of timber loss occurs during timber extraction, including tree felling in forest, logs to final useable products conversion, and manufacture of final solid wood products [4]. Limited studies have been carried out regarding harvesting losses during logging in mature forest stands [5]. For higher timber recovery, harvesting trees and their processing with minimal timber loss is essential for local and national economic growth [6]. Harvesting loss rate during logging varies in different studies depending on its local conditions, often considered 1:1 (1 m³ of extracted logs results in 1 m³ of residue) as the rule of thumb [7]. Many studies concluded a wide range of timber loss during selective logging, i.e., one to five times the extracted timber, indicating a recovery rate starting from 20 percent [8].

Logging activities have increased rapidly in recent years, wherein a wide use of high impact conventional logging has led to forest damage and fragmentation [5]. Conventional logging (CL) includes unplanned, merchantable harvest of size stems, felled by fellers and transported

by tractors or skid by skidders to depot [9]. Conventional logging results in neighboring tree damage and regeneration loss, breaks hydrological cycles, and enhances erosion in sloppy areas [10]. Instead of the CL, reduced impact logging (RIL) is gaining more attention in recent decades due to unplanned logging practices, safety concern of less workers, and higher timber and regeneration loss [11]. Even though the RIL is not a new concept, it includes good harvest and operational planning that can minimize harvesting losses and increase timber recovery with safer harvesting operations [9]. In the tropics, it has been proven that RIL is far better than conventional logging [10,12]. In the context of Nepal, conventional logging practices are being applied to harvest trees throughout the country [13].

The forest of Nepal has a minimum timber production potential of 1.66 million m$^3$, having the minimum employment potential of 400,000 full-time jobs per year [14]. Unfortunately, it has been providing only 0.5 million m$^3$ and nearly 45,000 full-time jobs annually with existing timber production practices [15]. The timber demand and supply scenario shows that Nepal has been facing a shortage of timber and fuel wood for many years. Timber, particularly round log, imports exceed the exports of wood-based timber product, which drains a significant amount of foreign currency and contributes to the trade deficits of the country. In 2018, Nepal imported 727,106 m$^3$ of logs and 514,472 m$^3$ in the year 2019 [16]. For the fiscal year 2018/2019, timber imports were worth USD 38 million, which was 65 times higher compared to the export of timber products [17]. The lack of appropriate forest management practices undermines the realization of the full potential of forests, resulting in an annual loss of USD 91 million [18]. The underperformance of community forests and underutilization of over matured *Terai* forests are the major causes for timber scarcity in Nepal [19].

In community-based forest management (CBFM), harvesting activities are carried out by community forest user group (CFUG) themselves or by hired contractors in close supervision of the respective CFUG and government forest officers [20]. The harvesting is semi-mechanized, whereby conventional loggers use chainsaws and traditional axes for felling, limbing, and bucking of trees. Mostly, band saws are being used for log sawing by a conventional sawyer in sawmills. Two-man felling saws and cross-cut saws are still being used, particularly in mountainous regions. Likewise, farm tractors are the most preferred means for skidding in the forests and transporting timber from the forests to the depot areas. Mechanized harvesting has not yet been practiced in the country [21].

Processing those round logs with minimum loss is one of the most important tasks for timber producers. Generally, the volume obtained from forest inventories indicates the gross volume of standing trees [22]. As a result of increasing demand for forest products, the price of round logs, sawn wood, and wood products has increased rapidly in Nepal [23].

Different rates of timber recovery or losses between different studies in different countries vary due to quality of logs, different sawing equipment used, and the final aim of product size [24]. According to Köhl and colleagues [22], harvesting studies are crucial for planning and for the inspection of harvesting processes that recognize economic and ecological unsustainable activities such as illegal logging and overexploitation.

Although harvesting loss is a serious issue in Nepal, no studies have been conducted to date. The dearth of information about timber losses occurring in each stage of tree conversion indicates unplanned harvest by less skilled manpower. *Shorea robusta* timber is recognized as one of the high commercial value timber species in Nepal. However, the loss of *Shorea robusta* timber in different stages of tree harvesting processes is being ignored. The loss has reduced state and local revenues. In this paper, we attempted to address the knowledge gap through the detailed assessment of the objectives on volume reduction factor and loss reduction strategies for *Shorea robusta* harvesting in the *Terai* region of Nepal. We explored three main questions: (1) What is the volume change ratio from standing tree to the felled logs? (2) What is the volume change ratio from felled logs to utilized timber from the logs? (3) What are the corrective actions to minimize timber losses at each stage of *Shorea robusta* harvesting? To answer the research questions, we applied a framework, suggested by Köhl and colleagues [22], that assesses the loss from gross standing tree volume to outturn volume

and estimates timber recovery rate (Figure 2). During harvesting activities, only a part of the gross volume is extracted, which is known as drain. The remaining volume inside the forests is termed as logging losses. In general practice, the net volume is often computed through rough deductions from the gross volume, and the deductions are often subjective [22]. Harvesting study results in obtaining objective volume reduction factors for particular trees, tree sizes, species, or tree species groups [22]. For Nepal, this study would serve as a baseline study that fills the knowledge gap on timber loss along the timber production chain and suggests corrective actions to adopt for its control.

## 2. Materials and Methods

### 2.1. Studied Species

*Shorea robusta* is the sole dominant species in the *Terai Shorea robusta* forest of Nepal and is one of the most important timber species for subsistence and commercial purposes [25]. *Shorea robusta* covers a stem volume of about 15.7% of the country, representing around 45% of the *Terai* forest of Nepal, which is also the highest forest type cover in Nepal according to single species composition [26]. In terms of total stem volume, *Shorea robusta* is the main tree species of Nepal, consisting of 28.2% of the total stem volume representing 54.8% of the total stem volume of the *Terai* region of Nepal [27]. The sawn timber from *Shorea robusta* is one of the most expensive timbers in Nepal and Southeast Asian markets [28].

### 2.2. Study Area

Data were collected from two *Shorea robusta*-dominated forests and three sawmills in the Morang district (Figure 1). Two forests, namely, Sukuna community forest of harvested area of 3.23 hectares and Pathare sanischare collaborative forest of harvested area of 23.14 hectares were selected. The forests were selected on the basis of forest characteristics, i.e., late successional climatic climax vegetation [25]; same management practices; and similar harvesting equipment and techniques used in timber harvesting, i.e., power chain saw with the same specifications and the CL, respectively. Three sawmills, namely, Nitesh Aara sawmill (A), Birat sawmill (B), and Om Shakti sawmill (C), were selected. Sawmills were selected on the basis of similar sawing equipment used, i.e., a horizontal band saw with a simple manual carriage and a vertical band saw for re-sawing. The selected sawmills were semi-mechanized, being served by conventional workers.

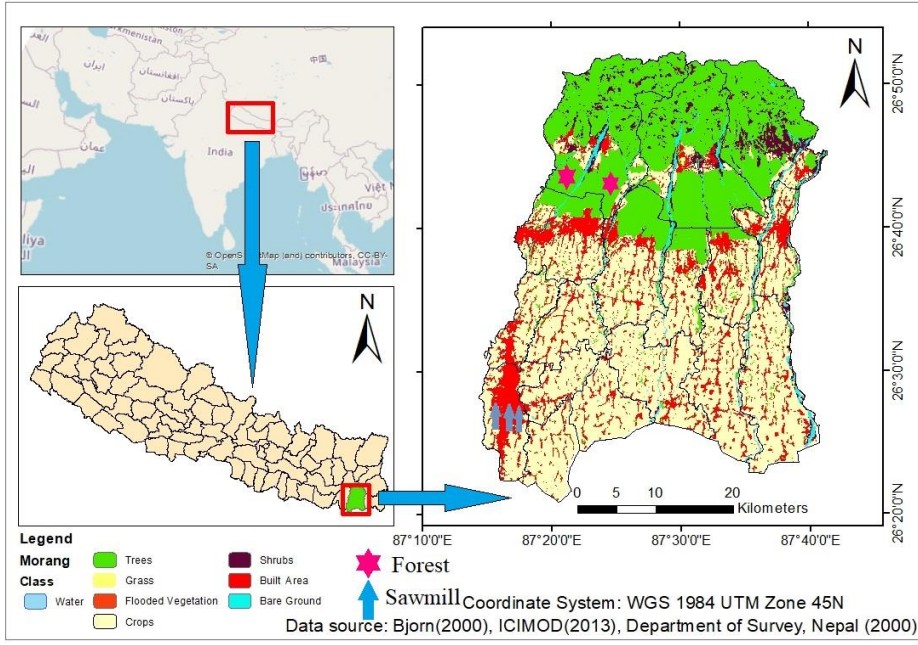

**Figure 1.** Map showing study area including the two sampled forests and three sampled sawmills.

### 2.3. Data Collection

Empirically, we studied quantification of timber losses by using the flow chart of Figure 2. Timber loss causes and reduction strategies were taken from expert interviews.

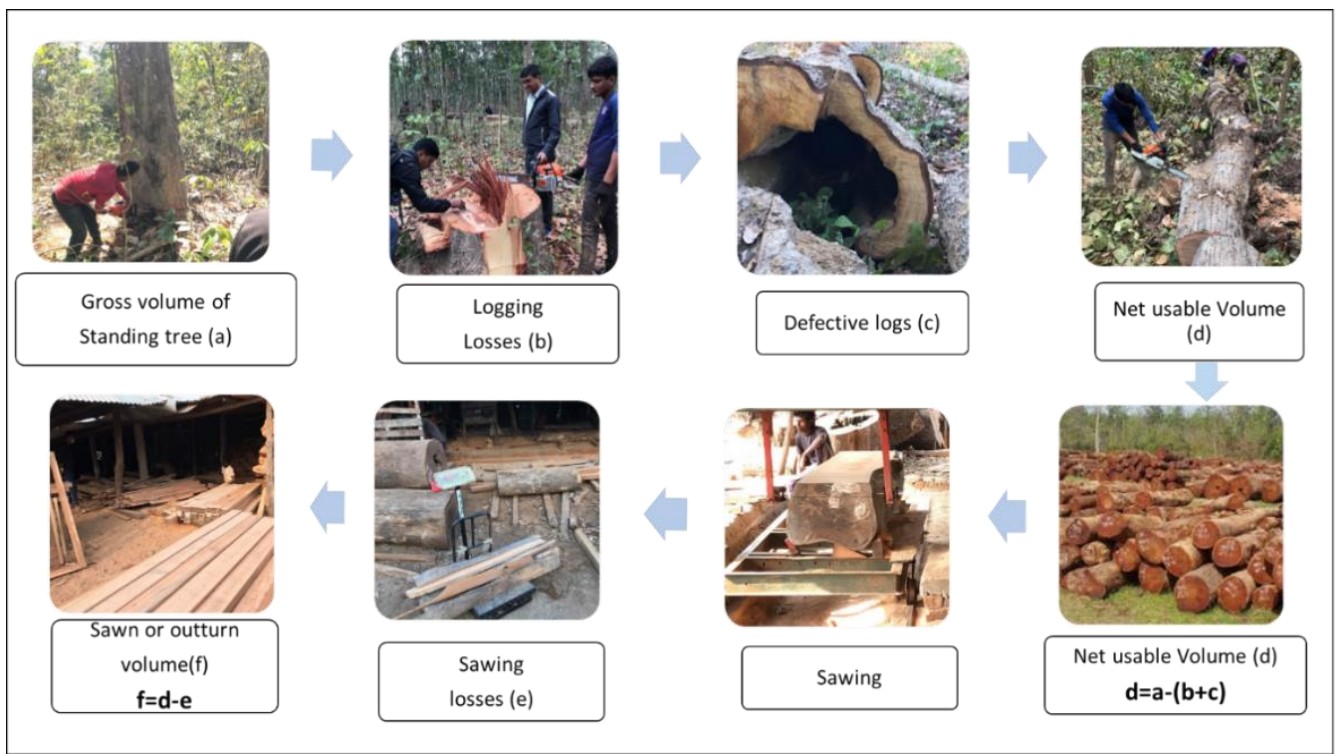

**Figure 2.** A schematic overview of research framework showing harvesting flow from standing tree to final outturn.

### 2.3.1. Measurements at Felling Sites

In total, 51 *Shorea robusta* trees harvested were from the felling coupe of both forests, i.e., 26 trees from Sakuna CF (Forest 1) and 25 trees from Pathari Sanischare CFM (Forest 2) were taken as sample trees in the month of March and April 2020. For the standing trees, diameter at breast height (DBH) at 1.3 m tree height was measured. The estimates of total tree height (H) and the tree bole height (BH) at first circular branching of the stem were taken. The DBH and tree total height were used to determine the standing volume of an individual tree [29,30]. Geographic coordinates, tree number, crown radii at four cardinal directions, and damage types of the marked trees were recorded for detailed information and identification of sampled trees after felling. After trees were felled, actual height and bole height were measured. Then, felled trees were grouped into desirable log size. The log conversion was performed by the tree fellers as their routine work. Length and over bark diameters at three positions of each log (upper end ($d_1$), middle of the log ($d_m$), and lower end ($d_2$)) were measured. Standing tree total height and tree bole height estimation were conducted using Apresys portable laser Rangefinder 5–550 M pro. Diameter at breast height, stump mid-diameter, and different stem diameters of logs were measured using diameter tape, whereas felled tree actual height, bole height, stump height, stem rot dimensions, and logs length were measured using linear tape.

### 2.3.2. Measurements at Sawmills

In total, 167 randomly selected logs (87 logs from sawmill A, 59 logs from sawmill B, and 21 logs from sawmill C) were assessed. Length and over bark diameters at three sections $d_1$, $d_m$, and $d_2$ of each log were measured. Log number, damages on the log, plausible reasons for the damages, and stem rot with its dimensions were measured and

noted. After sawing each log and converting it into different dimensions of sawn wood (e.g., planks), we individually recorded the quantity and the dimensions of the sawn wood (i.e., length, breadth, and height). Different diameters of logs were measured using diameter tape, whereas length of logs; length, breadth, and thickness of planks; and stem rot dimensions were measured using linear tape.

### 2.3.3. Questionnaire Survey

A semi-structured questionnaire survey was carried out on the basis of primary data collected from *Shorea robusta* tree felling and log sawing measurements. Samples were selected using expert purposive sampling. Government forest officials, CFUG and CFM members, and saw millers with more than 3 years of experience in *Shorea robusta* harvesting and processing activities were used as forest experts for this study. Experts were requested to fill out the online forms created in Google to share their views and experiences regarding harvesting losses in different harvesting stages, plausible causes for the losses, and strategies to reduce the losses.

### 2.4. Data Analysis

All fata from field measurements and questionnaire survey were analyzed using R studio, and paired *t*-test was performed.

### 2.4.1. Comparison: Estimated Standing Volume, Felled Volume, and Log Volume

The standing volume of a standing tree was estimated using an allometric equation that requires DBH and the total tree height of a tree (Table 1). As recommended by [26], coefficients a, b, and c of *Shorea robusta* species were taken as −2.4554, 1.9026, and 0.8352, respectively. Volume of the felled tree was also calculated using the same equation with measured tree attributes: DBH and measured total height of the tree. Stem rot volume was calculated using the length, breadth, and height of the rotted part (Table 1). Newton's formula was applied for the volume calculation of logs because of its greater accuracy [31]. The stem rot of each log was also calculated using the same equation for a felled tree. The recovered volume was analyzed using the volume recovery index (VRI) as the ratio of extracted volume to actual volume [32] and presented in quantity and percentage. Paired *t*-test was performed to understand whether there was significant loss in tree felling and sawing stages or not.

**Table 1.** Table showing equations used to estimate different volumes.

| S.N. | Estimates | Model Used | Description of Notations |
|---|---|---|---|
| 1. | Standing and felled tree volume | $L_n(V) = a + bLn\,(d) + cLn\,(h)$ Where $a = -2.4554$, $b = 1.9026$, and $c = 0.8352$ (Sharma and Pukkala, 1990) [33] | $L_n$ = natural log base; $V$ = volume in m$^3$; $d$ = diameter at breast height in cm; $h$ = total tree height in m; $a$, $b$, and $c$ are coefficients of species. ***Note**: values were divided by 1000 to convert them to m$^3$* |
| 2. | Stump volume | $V_s = \pi d^2 \times h/4$ | $Vs$ = stump volume in m$^3$, $d$ = mid-height diameter of the stump in m, and $h$ = stump height in m |
| 3. | Stem rot and sawn planks volume | $V = l \times b \times h$ | $V$ = volume in m$^3$, $l$ = length in m, $b$ = breadth in m, and $h$ = height in m |
| 4. | Log volume | $V = (S_1 + 4\,S_m + S_2) \times L/6$ (Newton's formula) | $V$ = log volume in m$^3$, $S_1$ = upper-end basal area in m$^2$, $S_m$ = middle basal area in m$^2$, $S_2$ = lower-end basal area in m$^2$, $L$ = length of the log in m, and basal area $(S) = \pi d^2/4$ in m$^2$ |

2.4.2. Quantification of Losses in Different Stages of Harvesting

The volume of the standing trees, felled trees, and logs were determined. In this study, all the solid material other than timber was evaluated as losses. However, some of these solid materials might have other uses. Timber loss volume during felling and bucking was calculated by deducting the total timber volume of logs of a tree from the timber volume of the standing tree. Similarly, timber loss volume during sawing was calculated by deducting the total volume of planks obtained from the total volume of log sawed.

**3. Results**

*3.1. Growth Parameters and Estimated Volume*

Table 2 below presents a basic statistical description of a subset of data, i.e., tree DBH and middle diameter of logs. Total estimated volume is the total timber volume of the sampled trees and logs.

**Table 2.** Table showing growth parameters and estimated volume of sampled trees and logs.

| Type | No. | DBH/$d_m$ (cm) * | | | | | Height/Length (m) ** | | | | | Total Estimated Volume (m³) *** |
|------|-----|------|-----|-----|----------|------|------|------|------|----------|------|------|
| | | Mean | Min | Max | S.D. (σ) | S.E. | Mean | Min | Max | S.D. (σ) | S.E. | |
| Tree | 51 | 72.9 | 47 | 96 | 12.29 | 1.72 | 30.78 | 15.20 | 48.90 | 5.84 | 0.82 | 299.11 |
| Log | 167 | 57 | 32 | 79 | 0.11 | 0.01 | 1.99 | 1.52 | 2.74 | 0.27 | 0.02 | 89.98 |

\* DBH = diameter at breast height of a standing tree in cm measured over bark taken at the height of 1.3 m from the ground, and $d_m$ is over bark mid diameter of logs. \*\* Height/length = total height of the standing tree and log length in m. \*\*\* Total over bark volume of the standing tree in m³ using the allometric equation ($a + b \times ln$ (DBH) $+ c \times ln$ (H))/1000) and logs in m³ using Newton's volume calculation formula ($S_1 + 4S_m + S_2$) $\times L/6$.

*3.2. Difference between Standing Tree Volume, Felled Tree Volume, and Log Volume*

Table 3 presents the total volumes estimated for standing trees, felled-tree volume, and log volume ($n = 51$ trees). The timber loss rates during conversion from felled tree to logs with and without stem rot were 23.39% and 21.59%, respectively. Paired *t*-test at the 5% level of significance revealed that there was significant timber loss in tree felling stage with the present felling practice (*p*-value = $8.186 \times 10^{-10}$).

**Table 3.** Table showing total volumes of standing trees, felled trees, bucked logs, and timber loss.

| Estimated Standing Tree Volume (m³) | Actual Felled Tree Volume (m³) | Bucked Logs Volume (m³) | Bucked Logs Volume after Deducting Stem Rot (m³) | Timber Loss (%) | Timber Loss after Deducting Stem Rot (%) |
|------|------|------|------|------|------|
| (*a*) | (*b*) | (*c*) | (*d*) | (*e*) = (($b - c$)/$b$) $\times$ 100 | (*f*) = (($b - d$)/$b$) $\times$ 100 |
| 299.1 | 277.11 | 217.29 | 212.3 | 21.59 | 23.38 |

Felled tree volume was reduced to 277.11 m³ from the standing tree volume of 299.11 m³. The reduction in volume was 7.4%. The bucked log volume was reduced by 21.6% from felled volume (Figure 3). Bucked log volume was considered as a commercial volume. Furthermore, the bucked volume was reduced by an additional 1.8% when the volume reduced by stem rot was deducted.

The result of the study indicates that a considerable portion of the total timber loss in this stage was associated with decay of heart wood. Out of 51 sampled trees, 27 (53%) had stem rot, whereas 3 of the trees were completely rotten. The percent of the stem volume infected by the heart wood rot ranged widely among trees with stem rot (2.43% to 100%). The loss averaged 7.58% across all trees felled.

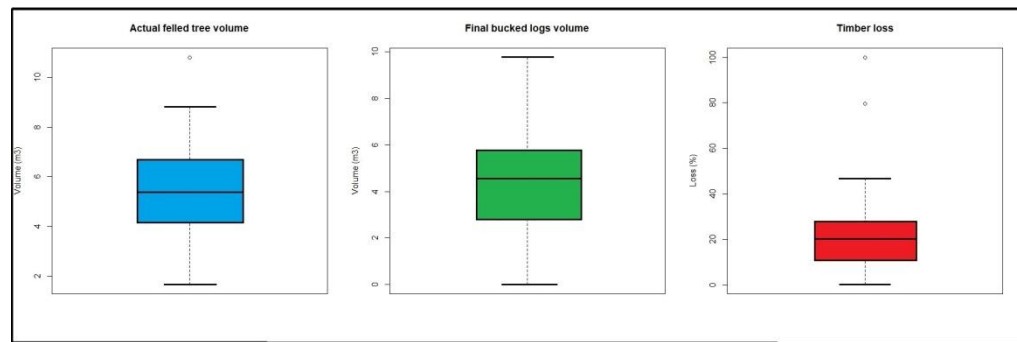

**Figure 3.** Box plots showing variations in actual felled tree volume, final bucked logs, and timber losses in tree felling stage.

### 3.3. Difference between Felled Log Volume and Utilized Timber Volume

Table 4 presents the volumes of actual felled logs, final outturn, and stem rot (*n* = 167 logs). The final outturn volume was reduced by 30.8% from log volume (Figure 4). Paired *t*-test at 5% level of significance revealed that there was significant timber loss in log sawing stage with the present felling practice (*p*-value = $2.2 \times 10^{-16}$).

**Table 4.** Volumes of actual felled logs, final outturn, stem rot, and timber loss.

| Total Log Volume (m³) | Log Volume after Deducting Stem Rot (m³) | Final Outturn Volume (m³) | Stem Rot Volume (m³) | Timber Loss (%) | Timber Loss after Deducting Stem Rot (%) |
|---|---|---|---|---|---|
| (a) | (b) | (c) | $d = (a - b)$ | $e = (a - c)/a \times 100$ | $f = (b - c)/b \times 100$? |
| 89.98 | 88.59 | 62.26 | 1.39 | 30.81 | 29.72 |

The results indicate that a considerable portion of the timber loss was associated with decay of heart wood. Out of 167 sampled logs, 33 (19.8%) had stem rot. The percent of the stem volume infected by the heart wood rot ranged widely among trees with stem rot (2.2% to 48.1%).

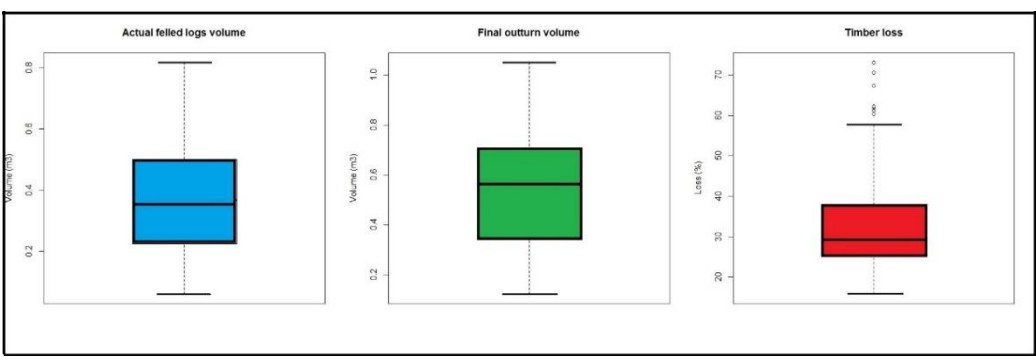

**Figure 4.** Box plots showing variations in actual felled log volume, final outturn volume, and timber losses in log sawing stage.

### 3.4. Timber Loss Reduction Strategies

Out of 132 forest experts, 116 (i.e., 87.8%) responses were received. Among them, 86 were government forest professionals, 19 were forest user group members, and 11 of them were saw millers.

### 3.4.1. Causes of Timber Loss

Experts were requested to provide their opinions regarding the harvesting loss of *Shorea robusta* timber. Since the harvesting practice of *Shorea robusta* is similar to other commercial species, for example, *Terminalia alata*, the same opinions remain valid. According to the experts, the major causes of losses occurring in all of the three harvesting stages (felling, bucking, and sawing) were the use of inappropriate equipment, followed by use of less-skilled manpower and poor ergonomic conditions (condition of the felling site, worker's safety and motivation) (Figure 5).

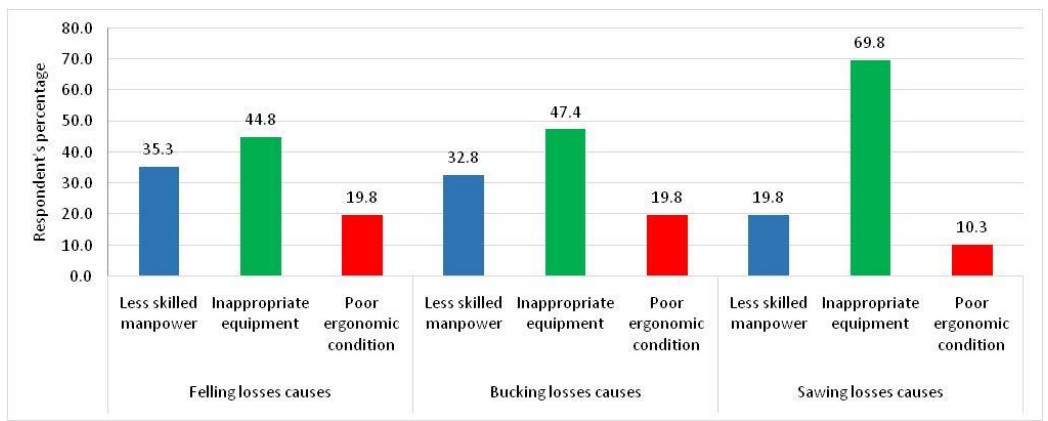

**Figure 5.** Bar graph showing causes of losses in different stages of the timber harvesting process.

### 3.4.2. Timber Losses Reduction Strategies

Introducing or promoting the appropriate harvesting equipment was the most prioritized strategy suggested by the experts in order to minimize timber losses in all of the three stages of timber harvesting. Training on RIL for felling and bucking losses reduction and enhanced ergonomic condition for sawing loss reduction were the other two highly recommended strategies (Figure 6).

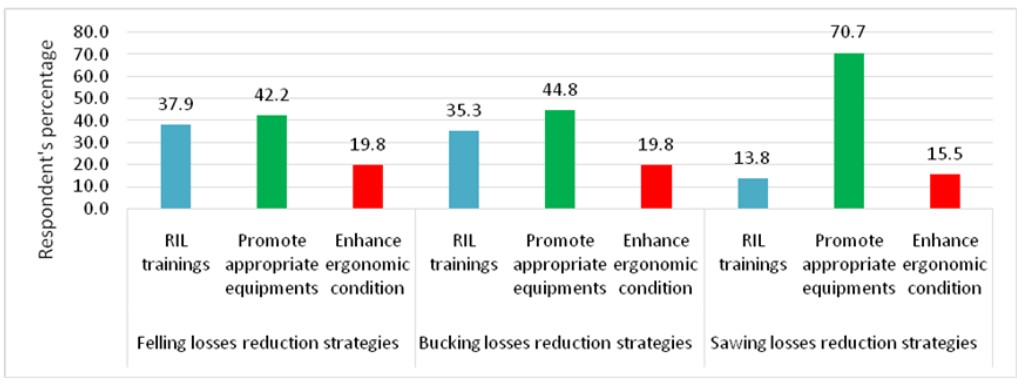

**Figure 6.** Bar graph showing major strategies suggested by the forest experts in order to reduce the timber losses along with the different stages of the timber harvesting process.

## 4. Discussion

### 4.1. Differences on Standing Tree Volume, Felled Tree Volume, and Log Volume

The findings from this study indicate that at least one-fifth of stem volume losses occurred during the felling and bucking stages of *Shorea robusta* timber harvesting. The finding corroborates with other studies conducted in similar climatic domains. However, more variations in tree harvesting loss rate can be seen in different studies (Table 5).

**Table 5.** Table showing tree harvesting loss rate of previous studies.

| Study Area | Loss Rate | Authors |
|---|---|---|
| Nepal | 19.8% (left over) | [34] |
| Nepal | 27% | [35] |
| Gabon | 25% | [36] |
| Ghana | 30% | [37] |
| Latin America | 44% | Dykstra and Heinrich [34] * |
| Africa | 46% | Dykstra and Heinrich [34] * |
| Sarawak Malaysia | 46% | Noack [34] * |
| Australia | 47.20% | [34] |
| Asia | 50% (1:1 ratio) | [8] |
| Tropical region (avg.) | 50% | Dykstra and Heinrich [34] * |
| Asia–Pacific | 54% | Dykstra and Heinrich [34] * |
| The Philippines | 60% | [38] |
| Brazilian Amazon | 66% (1:2 ratio) | [39] |
| *Terai Shorea robusta* forest, Nepal | 21.59% | This study |
| | 23.58% (after deducting stem rot) | |

* Cited in [34].

The key factors for these losses were associated with stem rot, higher stump, and other logging losses left in the forest after harvesting. Harvesting loss rate during logging varies in different studies depending on its local conditions, often considered 1:1 (i.e., 1 $m^3$ of extracted logs result in 1 $m^3$ of logging residue) as a thumb rule [7]. Many scholars indicated a wide range of timber loss during selective logging, i.e., one to five times the extracted timber, indicating a recovery rate starting from 20% [8].

The logging loss rate estimated in this study is similar to the results of [37] in Ghana and [36] in Gabon. Both studies were conducted in the tropical forest, adopting a similar methodological framework as in this study. The rate of timber loss found in this study corroborates with the study by [35] in which the logging loss rate was 27% of total timber volume in the *Terai* region of Nepal. Compared to our study, Poudyal and colleagues [34], who carried out research in the same region of this study area, found 19.8% of timber loss. The lower loss rate could have been due to multiple species timber loss estimation and exclusion of completely rotted (damaged) trees in the analysis.

Table 5 shows that most of the studies indicated that around half of the stem volume is lost during timber harvesting. The rates are considerably high as compared to our findings. Plausible reasons for the higher rate of loss could be (i) the studies considered total tree volume including branch volume, whereas this study considered stem/timber volume only as the total volume of the tree, i.e., clean bole volume of a tree; (ii) the studies included transportation losses, which were excluded in our study; and (iii) the recovery rate calculation in the studies considered multiple species (i.e., average of recovery rate of the timber volume of multiple species), whereas we only studied the most economically valuable species in Nepal.

*4.2. Differences on Felled Log Volume and Outturn Volume*

Our results indicate that nearly one-third of the total timber volume is reduced while converting the round logs into sawn wood by sawing process (Table 6). Those residues were caused by sawdust, *Bakal* (unfit and chips), stem rot, knots, bark removal, and side slabs. Stem rot reduces sawn-timber volume considerably. Difference in rates of sawn-timber recovery and residues between different studies and countries could be due to different sawing equipment, quality of logs, and end-product size [24].

**Table 6.** Rates of timber loss while converting round logs into sawn-wood as estimated by timber utilization studies conducted in various countries/regions.

| Study Area | Loss Rate | References |
|---|---|---|
| Indonesia | 20–30% | [3] |
| Nepal | 39% | [34] |
| China | 40% | Chen [40,41] * |
| Nigeria | 43.92% | [42] |
| Malaysia | 45% | Poyry [40,41] * |
| Indonesia | 46% | Gintings and Roliadi [40,41] * |
| Malaysia | 48% | Rayn [40,41] * |
| Many developing countries | 49.20 (40–58)% | [41] |
| Asia–Pacific | 50% | Dykstra and Heinrich [40,41] * |
| Southeast Asia | 50% | IUFRO [40,41] * |
| Papua New Guinea | 50% | FAO [40,41]* |
| Malaysia | 50–58% | [7] |
| *Terai Shorea robusta* forest, Nepal | 30.81% | This study |
| | 29.72% (after deducting stem rot) | |

* Cited by [40,41].

Sawn timber loss rate of this study was lower than several other studies. Koopsman and Koppejan estimated a timber loss of 50–58% while converting the round logs to sawn-timber [7]. Similarly, Enters reported average saw mill loss rates of 49.2% (40–58%) in many developing countries [41] (Table 6). The high recovery rate in Nepal (this study) compared to Koopsman and Koppejan [7] and Enters [41] could be due to most of the end-products of *Shorea robusta* logs being used primarily for building construction, wherein the quality of *Shorea robusta* logs and their sizes used are more flexible in Nepal [13]. Sawing timber loss rate of this study was also lower than many other studies (Table 6). Reasons could be (i) *Shorea robusta* being the strongest and valuable timber species in Nepal, being able to hold gum and nails properly—this implies that even a small size of timber (i.e., 2.54 cm × 2.54 cm) was also taken as a useable product for furniture and finishing wood works, which result in a higher timber recovery rate, and (ii) exclusion of timber loss during the conversion of planks to end-use products. In this study, only sawn planks from *Shorea robusta* logs were considered as final outturn, excluding timber losses from further processing of the sawn planks into end-use products such as doors, windows, and any other furniture products.

In contrast, a study from Malik and Hopewell [3] with similar methods and limitations concluded that the sawing loss rate was 20–30%, which is slightly less than our study. This could have been due to (i) study area: Jepara, Indonesia (mechanized furniture production with advanced equipment used for sawing); (ii) live sawing pattern in small green logs adopted, which generally results in higher green-off saw recovery rates; and (iii) targeted final products were for chairs or table legs and tops, which could also use small pieces of sawn wood.

*4.3. Timber Loss Reduction Strategies*

The majority of the forest experts expressed that the leading causes for harvesting loss in all of the three stages (i.e., felling, bucking, and sawing) of timber harvesting were the use of inappropriate equipment, unskilled or less skilled manpower, and poor ergonomic conditions. Eroğlu and colleagues [43] suggested that an appropriate choice of tools and technology during tree harvesting can lead to optimum timber production with minimal losses. Traditional harvesting practice is the leading cause for high log damage during forest tree harvesting [44]. The use of conventional saw mills results in high amounts of timber loss during transportation and sawing in comparison of using portable saw mills in felling sites. Such losses could be reduced considerably by using flexible and portable sawmills in the felling sites [45]. Almost all of the fellers, buckers, and sawyers fall into

categories of unskilled and semi-skilled laborers. Thus far, in Nepal, there is neither a designated institution nor a dedicated course to provide the loggers with the training required to implement improved timber harvesting practices. Apart from that, no license is required to operate harvesting equipment such as chainsaws, which are easily available in markets and can be purchased by any interested individual (information obtained from power chainsaw owners and tree fellers in the study areas). Setting minimum standards for harvesting operation including specifying required skills for loggers and provision of harvesting trainings for them are crucial for effective and efficient harvesting in all of the timber-producing countries [46].

Most of the forest experts believed that promotion and use of appropriate equipment for tree felling and bucking can considerably contribute to the reduction of felling and bucking losses. In Nepal, manually operating a chainsaw is commonly used for felling and bucking of trees. The choice of tools and technology of work makes an impact on the degree of damage to residual trees, regeneration, and assortment of harvesting activities [43]. Mechanized harvesting tends to cause less damage than using a chainsaw for felling and bucking trees [47]. Vanderberg and colleagues [44] also concluded that the most valued losses during harvesting of trees were caused by a manual harvesting system. Felling and bucking can be done manually, motor-manually, or by mechanized harvesters and feller bunchers [48]. The majority of productive *Shorea robusta* forests of Nepal are located in flat land that can easily accessed by mechanized harvesters. However, adequate capacity and investment are needed for such a harvesting option, and the volume of harvest needs to be assessed thoroughly before introducing such machinery. In this context, introduction of semi-mechanized equipment such as powerful chainsaws for felling and strong synthetic ropes for directional felling as well as skidding along with portable winches and wedges might reduce timber and economic losses substantially. Additionally, provision of required ergonomic conditions at the workplace minimizes risk and enhances timber recovery. More experienced and trained workers are able to minimize such losses by reducing the damages on the tree itself and damage to the residual stand during logging [49]. A team of highly trained/skilled buckers can control bucking losses; therefore, using such a team of skilled buckers moving from operation to operation is the best option [50].

For felling and bucking, training on RIL for both forest managers and forest workers was suggested by the forest experts as a second priority to minimize the timber losses. Various studies concluded that 30 to 50 percent damage to a stand can be reduced by using RIL techniques [10,12,51,52]. RIL is not only a technique to reduce the damage, but also a procedure to optimize resource utilization through forest inventory and planning of harvesting [53]. The STREAK project demonstrated that the RIL method reduced logging damage by 50% in comparison with CL in Eastern Kalimanthan [53]. The level of cut in RIL is generally lower than in CLs [54]. One of the constraints to implement RIL is it requires high management expense for more supervision, planning, and training for managers and workers [53]. RIL demands substantial resources (financial, human and logistics), and sometimes is also criticized as "reduced income logging" for a few years in the beginning.

In this study, the majority of forest experts suggested promotion of appropriate equipment as the best strategy to reduce sawing loss. All of the three sawmills in the study were similar in technology, consisting of a horizontal band saw with a very simple manual carriage and a vertical band saw for re-sawing. All of the sawmills used a manually pushing system for log transfer into sawing frame and sawing into wood pieces. Small, portable, and flexible sawmills are better than conventional sawmills in terms of timber recovery rate and less damage to stand [45]. Small portable sawmills are environmentally friendly, highly profitable, and easy to operate by local forest users. Sargent and Burgess [55] concluded that small portable sawmills have the potential for sustainable harvesting of natural forests. By using portable sawmills in the forest, losses occurred during transportation, and skidding can be minimized, which can restrict over logging. Performing sawing activities in a felling site using small portable sawmills can reduce left-overs in the forest [56]. Olufemi and colleagues [42] suggested that the introduction of advanced sawing practices and secondary

wood processing machines is the best option for timber loss reduction. Sawing practices are more time-consuming (approximately 1 h per log) and more demanding of manpower (4–5 manpower per log sawing) in Nepal. It could be improved by introducing flexible and portable sawmills in felling sites or mechanized loading and log sawing machines in fixed sawmills, which could result in lower timber losses per log sawn. Small portable sawmills require relatively low establishment costs and are less labor-intensive. In the case of Nepal, such mills could be very attractive and relevant for CFUGs. A group of CFUGs can afford such mills and operate locally. In contrast, even though improved equipment and personnel are used in sawmills, traditional practice and lack of secondary slab waste processing machines are the major reasons for high sawing product loss [42]. Malik and Hopewell [3] suggested that live sawing patterns can be an appropriate method for reducing sawing losses. This method works well in green logs and small furniture products making logs. However, it may not fit in the case of semi-dry to dry *Shorea robusta* logs that have high wood density. This may require long sawing time, which implies more labor and power costs as well as leading to more losses.

Even though the response from forest experts put ergonomic context as the lowest priority on both timber loss cause and reduction strategy, it cannot be ignored completely. The reasons for this low priority are not only due to the lack of human and financial resources, but also a general belief that expenses in this field area are more related to liability rather than the profitable investment. Ergonomics aimed at reducing the costs associated with equipment downtime, underutilization/overexploitation of forest resources, suboptimal processing capacity, and other related problems that result in high long-term benefits [57]. Putz and colleagues [58] concluded that the world's most dangerous occupation also includes tropical forest logging (e.g., from 1976 to 1989, 5–10 deaths per million m$^3$ harvested reported in Sarawak, Malaysia). Both workers and managers require more safety training and also require motivation first to improve their harvesting skills [58].

## 5. Final Remarks and Recommendations

In dipterocarp forest, conventional logging practices where the shelterwood systems with subsequent regeneration felling are being practiced result in higher timber loss and other damages on the forest stand. In natural forest harvesting practices using selective logging, a considerable amount of timber is lost while trees are felled in Nepal. Sawing logs in traditional sawmills is also associated with a huge timber loss. More than one-fifth of the timber volume loss occurs in the tree harvesting stage, whereas about one-third timber volume loss occurs in the sawing stage. Efforts put into reducing logging and sawing losses can increase economic potential through enhancement of timber recovery rate as well as the secondary use of the logging and sawmill residues. Reduction in logging losses can reduce forest degradation and enhance the contribution of the forest sector to mitigate climate change in the long term. This study serves as a baseline study to identify and quantify timber loss in different stages of tree conversion and also suggests some pragmatic reduction strategies. Timber recovery rate (standing volume, log volume, outturn volume) and information on possible damages and timber loss in particular stages of timber harvesting help forest owners in various ways. With the help of such harvesting or recovery study, for example, local forest user groups can estimate the outturn volume of usable timber for distribution, forest offices can prepare informed forest management and harvesting plans, and central/provincial government can prepare forest harvesting guidelines. Division Forest Office and CFUGs may adopt appropriate timber losses reduction strategies focusing on impact logging covering wide forest management regimes in the *Terai* region of Nepal.

**Author Contributions:** Conceptualization; U.A., P.R.N. and B.R.; methodology; U.A. and P.R.N.; software; U.A., P.R.N. and M.M.; formal analysis; U.A. and P.R.N.; data curation; U.A.; writing—original draft preparation; U.A.; writing—review and editing; U.A., P.R.N. and M.M.; visualization; U.A.; supervision; P.R.N. and M.M.; project management; U.A., P.R.N. and B.R. All authors have read and agreed to the published version of the manuscript.

**Funding:** This research received no external funding.

**Institutional Review Board Statement:** Not applicable.

**Informed Consent Statement:** Not applicable.

**Data Availability Statement:** The data used in this study will be available upon request from the first author.

**Acknowledgments:** Special thanks to Bishal Ghimire, Kumar Darjee, Ashok Parajuli, Shankar Tripathi, Surendra Ranpal, Kishor Aryal, Prashant Paudel, and Parmeshwor Paswan for providing assistance in data collection and suggestions in the writing phase. Our sincere thanks to Irene Wangari Mukure for language editing support.

**Conflicts of Interest:** The authors declare no conflict of interest.

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
