# Peer review of "Timber Losses during Harvesting in Managed Shorea robusta Forests of Nepal"

_land, doi:10.3390/land11010067_

Round 1
Reviewer 1 Report
General comment: The authors have done a good amount of work to present baseline information on the timber losses at different stages and formulate the reduction strategies.
However, some points are still needed to be considered before publication. For example, firstly, to restructure the paper's content. Secondly, to add a conceptual framework of the study that connected the quantification of timber losses and results of the expert interview.
Detail comments
- Page 2, paragraph 2: The forest area of Nepal has minimum forest products ….
Wood and non wood forest products?
- Figure 1 should be placed in Material and Methods because it explains the flowchart of quantification of timber losses. Thus, the sentences explained the Figure 1 should also be moved to the Methods.
- 3.3 Questionnaire survey: should explain only the methods of the survey, (e.g., methods of interview, type of questions asked that relevant for this article, sampling procedure, criteria of the respondents). Hence, the number of the final respondents and their characteristics should be placed in the Results section.
- 4. Data analysis: It is not clear why you used Paired T-test for the questionnaire survey. Did you ask some variables that provided before and after measurements/treatments to the experts? I suggested making a more detailed explanation of which data used the Paired T-test, and if any, which other data used other statistical analysis.
- Table 8 (Average/mean?)
- In 3.2 and 3.3. the authors stated that the cause of the timber losses was fungal decay. If it was not part of the Results, please move it to the discussions section and add a reference. If it was part of the results, please mention from which part of the study the author obtained the cause of fungal decay (e.g., based on the interview).
- Apparently, fungal decay was a significant cause of timber losses in this study. Thus, in the discussion, I suggest adding an explanation of how the fungus deteriorates the woods, and how to prevent it in the Timber losses reduction strategies.
- In the Conclusions, I suggested adding the fungus problems and preventions.
Author Response
Comment
General comment: The authors have done a good amount of work to present baseline information on the timber losses at different stages and formulate the reduction strategies.
However, some points are still needed to be considered before publication. For example, firstly, to restructure the paper's content. Secondly, to add a conceptual framework of the study that connected the quantification of timber losses and results of the expert interview.
Response
Thank you for the suggestions. We worked in manuscript and incorporated your suggestions. We shift framework from timber losses quantification from introduction to material and methods section. Also, we shifted respondent's selection and criteria from material and methods section to result section. We changed and restructure the paper's content as suggested.
Framework for timber losses quantification was taken from fieldwork whereas; cause and reduction strategies were identified by using expert interview. Empirically, we studied quantification of timber losses by using flow chart in figure 2. Causes and reduction strategies were taken from experts interviews. So, we add a line "Empirically, we studied quantification of timber losses by using flow chart of figure 2. Timber losses cause and reduction strategies were taken from experts interviews. " in section 2.3 as suggested.
Detail comments
Comment 1
Page 2, paragraph 2: The forest area of Nepal has minimum forest products ….
Wood and non wood forest products?
Response 1
Thank you for the suggestion. We changed it to timber production instead of forest products production. Now the line reads as " The forest of Nepal has minimum timber production potential of 1.66 million m3 having the minimum employment potential of 400,000 fulltime jobs per year".
Comment 2
Figure 1 should be placed in Material and Methods because it explains the flowchart of quantification of timber losses. Thus, the sentences explained the Figure 1 should also be moved to the Methods.
Response 2
Thank you for the suggestion. We put the flowchart of quantification of timber losses in Material and methods section as suggested. Also, we changed Figure 2, Map showing study area including sampled two forests and three sawmills. to Figure 1. Map showing study area including sampled two forests and three sawmills. Also, we changed figure 1 to figure 2 in 13th line of last paragraph of introduction section.
Comment 3
3.3 Questionnaire survey: should explain only the methods of the survey, (e.g., methods of interview, type of questions asked that relevant for this article, sampling procedure, criteria of the respondents). Hence, the number of the final respondents and their characteristics should be placed in the Results section.
Response 3
Thank you for the suggestion. We placed number of final respondents in result section. But, characteristics of respondents are one of our criteria or methods designed to select respondents. So, in our perception, it fits better in material and method section. So, we keep it as it is.
Comment 4
- Data analysis: It is not clear why you used Paired T-test for the questionnaire survey. Did you ask some variables that provided before and after measurements/treatments to the experts? I suggested making a more detailed explanation of which data used the Paired T-test, and if any, which other data used other statistical analysis.
Response 4
Thank you for your suggestion. We do not use paired T-test for the questionnaire survey. We used individual tree volume and timber loss volume for Paired T test during tree felling stage. Similarly, we used individual log volume and timber loss volume of each log for paired T test during sawing stage. We compared timber volume before and after felling or sawing. We used paired T test to test whether the timber losses during different stages are significant or not.
Comment 5
Table 8 (Average/mean?)
Response 5
Thank you for your query. These values in Table 8 are total logs volume. So we added word "Total" to make it clearer.
Comment 6
In 3.2 and 3.3.the authors stated that the cause of the timber losses was fungal decay. If it was not part of the Results, please move it to the discussions section and add a reference. If it was part of the results, please mention from which part of the study the author obtained the cause of fungal decay (e.g., based on the interview).
Response 6
Thank you for your suggestion. In both sections 3.2 and 3.3, we removed the word "fungal". It was our mistake that we termed as a fungal decay of heartwood. We just measured and quantified the stem rot or decay of heartwood during fieldwork. We do not investigate the causes of decay in samples. So as suggested we removed the word fungal.
Comment 7
Apparently, fungal decay was a significant cause of timber losses in this study. Thus, in the discussion, I suggest adding an explanation of how the fungus deteriorates the woods, and how to prevent it in the timber losses reduction strategies.
Response 7
Thank you for your suggestion. As suggested for further explanation of fungal decay and its details, we actually do not identify the cause of decay in trees. So, we do not discussion about fungal decay in this paper. Fungal decay is one of the causes of stem rot in Sal trees, so we termed it as fungal decay. But now we all authors discussed about it and finally concluded that we do not empirically investigate the cause of stem rot. Also, it was our mistake using word "fungal decay of heart wood". With your genuine suggestion, we recommended study need on "frequency and severity of stem rot with its causes" in conclusion and recommendations section. Now the line reads as " We recommend experimental studies focusing on, i) exploring and comparing appropriate harvesting equipment for felling, bucking and sawing taking all aspects (i.e. social, ecological and economical aspects) into consideration, ii) cost and benefit analysis of new equipment introduction, and iii) frequency and severity of stem rot with its causes and iv) reduced impact logging covering wide forest management regimes in the Terai region of Nepal."
Comment 8
In the Conclusions, I suggested adding the fungus problems and preventions.
Response 8
Thank you for your genuine feedback. Fungus problems and preventions are major topics to be studied. In this study, these are out of scope. We proposed in conclusion and recommendation section about study needed for frequency and severity of stem rot with its causes as mentioned in comment 8.

Reviewer 2 Report
> In context of Nepal,
In **the** context of Nepal,
> The forest area of Nepal has minimum forest products potential of 1.66 million m 3
> having the minimum employment potential of 400,000 fulltime jobs per year [14].
Where along the value chain?
> import exceeds the expert
export
> semi-skilled loggers use chainsaws and traditional axes for felling, limbing and
> bucking trees. Mostly, band saws are being used for logs sawing by semi-skilled
> manpower in saw mills.
"semi skilled" is pejorative, especially considering that the loss rates your report in
table 9 and 10 are lower than in many other countries. If the point is that operators
are not trained in RIL, you could call them CL loggers for example. Concerning saw mill
operators, you could remove the "semi-skilled" manpower in saw mills. It appears that
sawmill losses are not an issue related to the operators as later own you propose that
saw mill loss can mainly be improved by a change of machinery (portable sawmills).
> Mechanized harvesting has not practiced yet in the country
... has not been practiced
> Table 1 established date
What does this date mean? It cannot be the plantation date since these would be to
young. Please clarify this ambiguity.
> 2.3.1 Measurements at felling sites
Where the same logs re-measured between felling sites and sawmills?
Probably not, but the text left me wondering. Please clarify.
> Table 5
> Standing and felled tree volume L n (V) = a+bLn(d)+cLn(h)
What are the a, b, c coefficients used in the standing and felled tree volume
estimation?
> 2.4.3. Strategies to reduce harvesting losses
> Semi-structured questionnaires used in this study were constituted by open-ended and
> close-ended questions. Quantitative data obtained by the survey were analyzed us- ing
> descriptive statistics such as percentage, mean, frequency distribution and graphics
> whereas; qualitative data were illustrated using pie-charts, bar diagrams and tables.
These points about the questionnaires and data analysis where already mentioned before
or are self evident. Remove this paragraph, which is not about strategies. There is
already a section "3.4.2 Timber losses reduction strategies" later.
> Paired T test at 5% level of significance revealed that there is significant timber
> loss in tree felling stage with present felling practice (p-value = 8.186e-10).
This test is misleading, I suggest removing this sentence. What are the null and
alternative hypotheses? No one really doubts the existence or significance of losses. A
more interesting Hypothesis would be to test whether Loss_RIL are equal to Loss_CL or
whether the losses of RIL harvesting are significantly different. This would require
additional data obviously.
> Figure 3. Bar plots showing variations in actual felled tree volume, final bucked logs
> and timber losses in tree felling stage.
A scatter plot of losses would be more informative. For example you could plot losses on
the y axis, DBH on the x axis, using a different colour for the two harvest sites. You
could also provide the scatter plot with x=height, y=losses or x=volume, y=losses.
> Paired T test at 5% level of significance revealed that there is significant timber
> loss in logs sawing stage with present felling practice (p-value = 2.2e-16).
Is this test really needed? No one doubts the existence of saw mill losses. I would
advise you to remove this sentence.
> Figure 4. Bar plots showing variations in actual felled log volume, final outturn
> volume and timber losses in logs sawing stage.
To illustrate the variability across log diameters and companies, you could provide a
scatter plot of losses on the y axis and stump volume on the x axis, using a different
colour for the different saw mills and a different point form for the logs with stem
rot. Alternatively similar scatter plots could be made with x=basal area, y=losses or
x=Length of log, y=losses.
> Training on RIL for felling and bucking losses reduc- tion and enhanced ergonomic
> condition for sawing losses reduction were other two highly recommended strategies
> (Figure 6).
There are different aspects of RIL as you mentioned in the introduction (1) reduction of
damage to neighbouring trees and regeneration losses (2) increased timber recovery. You
insist on the second aspect, but the first aspect is probably the most relevant for long
term sustainable forest management. Coming back to this on my comment to section 4.3.
> The rates are considerably higher compared to our find- ings. Plausible reasons for
> the higher rate of loss could be: i) the studies considered total tree volume
> including branches volume, whereas this study considered stem/timber volume only as
> the total volume of tree i.e. clean bole volume of a tree, ii) the studies in- cluded
> transportation losses which was excluded in our study, and iii) the recovery rate
> calculation in the studies considered the multiple species (i.e. average of recovery
> rate of the timber volume of multiple species) whereas we studied only the
> economically most valuable species in Nepal.
Can you distinguish between coniferous and broad leaves? What about plantation forestry?
> Those residues were caused due to the saw dust, Bakal (unfit and chips), stem rot and
> knots.
Do you need to mention bark removal and side slabs?
> table 10 Terai Sal forest, Nepal 30.81% (with rot) 29.72% (excluding rot) This Study
Are these with/without rot loss rates consistent with the loss rates specified page 8 :
> "Out of 167 sampled logs, 33 (19.8%) had stem rot. The percent of the stem volume
> infected by the heart wood rot ranged widely among trees with stem rot (2.2% to
> 48.1%). The loss averaged 1.54% across all logs sawed."
Page 12
> chair or table legs and tops which could use outfit sawn wood too.
Clarify the meaning of "outfit sawn wood".
> The use of conventional saw mills results in **severe** timber losses during
> transportation and sawing
Temper this sentence since losses are actually lower than in many countries.
> Sal forests of Nepal are located in flat land which can easily accessed by mechanized
> harvesters.
Please mention the trade-off with soil damage potentially caused by mechanized
harvesters. Are harvester improving RIL mentioned in the introduction?
> RIL demands substantial resources (finan- cial, human and logistics), and sometimes is
> also criticized as ‘Reduced Income Logging’ for the few years in the beginning.
RIL can be more demanding on the harvesting companies. But forest owners typically need
to consider the whole growth cycle. Taking into account RIL's impacts on reneration and
neighboring trees. From an economic perspective, you could provide an estimation of the
discounted future value of the losses due to CL. Such value increase could improve
forest owner's financial incentive to perform RIL. RIL's improvement of future income
are not assessed in this study, but you could make a better attempt at providing
insights from the literature.
> By using portable saw mills in the forest, villagers will be able to decide about
> where and how much to fell which will **restrict over logging**.
How can a tool prevent over harvesting? I think this sentence is incorrect. There are
probably policies in place such as local or national forest codes that restrict over
harvesting. If yes, these regulations should be mentioned here as well as potential
enforcement issues.
> Ergonomics aimed at reducing the costs associated with equipment downtime,
> underutilization/overexploitation of forest resources,suboptimal processing capacity,
> and other related problems which results in higher long-term benefits
This sentence doesn't make sense, separate it in two or remove it.
> a **considerable** amount of timber is lost while felling trees in Nepal.
However losses are lower than in many other world regions according to table 9.
> Efforts paid in reducing logging and sawing losses can increase economic potential
> through enhancement of timber recovery rate as well as the secondary use of the
> logging and saw mill residues.
What is the evidence to support this statement?
The current study only studies CL. It would be interesting to compare CL and RIL in a
follow-up study.
Author Response
Comment 1
The forest area of Nepal has minimum forest products potential of 1.66 million m 3 having the minimum employment potential of 400,000 fulltime jobs per year [14].
Where along the value chain?
Response 1
Thank you for your comment. In the given line "Forest products" actually means "timber" and employment activities include tree felling, logging, sawing, transportation and furniture manufacture works". So we changed a line which reads as "The forest area of Nepal has minimum timber production potential of 1.66 million m3 having the minimum employment potential of 400,000 fulltime jobs per year including tree felling, logging, sawing, transportation and furniture manufacture works" [14].
Comment 2
import exceeds the expert
export
Response 2
Correction applied as suggested.
Comment 3
semi-skilled loggers use chainsaws and traditional axes for felling, limbing and bucking trees. Mostly, band saws are being used for logs sawing by semi-skilled manpower in saw mills.
"semi skilled" is pejorative, especially considering that the loss rates your report in table 9 and 10 are lower than in many other countries. If the point is that operators are not trained in RIL, you could call them CL loggers for example. Concerning saw milloperators, you could remove the "semi-skilled" manpower in saw mills. It appears thatsawmill losses are not an issue related to the operators as later own you propose thatsaw mill loss can mainly be improved by a change of machinery (portable sawmills).
Response 3
Thank you for the suggestions. We changed Semi skilled loggers to Conventional loggers throughout the manuscript as suggested.
Comment 4
Mechanized harvesting has not practiced yet in the country
... has not been practiced
Response 4
Correction applied as suggested.
Comment 5
Table 1 established date.
What does this date mean? It cannot be the plantation date since these would be toyoung. Please clarify this ambiguity.
Response 5
Thank you for the suggestion. Established date actually means forest handover date to Community forest user groups in Nepal. So we changed it to "Forest handover date" in table 1 as suggested.
Comment 6
2.3.1 Measurements at felling sites
Where the same logs re-measured between felling sites and sawmills? Probably not, but the text left me wondering. Please clarify.
Response 6
Thank you for your query. We took independent samples in felling site and saw mills, so we do not re-measured logs between felling site and saw mills. In Nepal, due to tender system to sell logs in Nepal, sampled logs obtained from felling site went to many saw mills so it was difficult to follow same logs. Therefore, we took independent samples in felling site and sawmills.
Comment 7
Table 5
Standing and felled tree volume L n (V) = a+bLn(d)+cLn(h); What are the a, b, c coefficients used in the standing and felled tree volumeestimation?
Response 7
Thank you for the suggestion. We added values of coefficients a=-2.4554, b=1.9025 and c=0.8352 as as suggested in table 5.
Comment 8
2.4.3. Strategies to reduce harvesting losses
Semi-structured questionnaires used in this study were constituted by open-ended andclose-ended questions. Quantitative data obtained by the survey were analyzed usingdescriptive statistics such as percentage, mean, frequency distribution and graphicswhereas; qualitative data were illustrated using pie-charts, bar diagrams and tables.
These points about the questionnaires and data analysis where already mentioned before or self evident. Remove this paragraph, which is not about strategies. There isalready a section "3.4.2 Timber losses reduction strategies" later.
Response 8
Thank you for the suggestion. We have removed section 2.4.3. as suggested.
Comment 9
Paired T test at 5% level of significance revealed that there is significant timber> loss in tree felling stage with present felling practice (p-value = 8.186e-10).
This test is misleading, I suggest removing this sentence. What are the null andalternative hypotheses? No one really doubts the existence or significance of losses. Amore interesting Hypothesis would be to test whether Loss RIL are equal to Loss CL or whether the losses of RIL harvesting are significantly different. This would requireadditional data obviously.
Response 9
Thank you for your suggestion. We took felling of trees and sawing of logs as an action oriented treatment. We took total volume of individual trees before applying treatment (felling or sawing) and volume of useable timber volume after applying treatment. We discussed within authors and with statistic experts about it. So, we used Paired T test. In this case, null hypothesis was a significant loss occurs during conventional felling of trees.
We only compared standing volume and outturn volume but not two different systems or not a case of RIL loss. So this is out of our scope. Your suggestions on comparison of loss during CL and RIL are very valuable and interesting.
Comment 10
Figure 3. Bar plots showing variations in actual felled tree volume, final bucked logsand timber losses in tree felling stage.
A scatter plot of losses would be more informative. For example, you could plot losses onthe y axis, DBH on the x axis, using a different colour for the two harvest sites. You could also provide the scatter plot with x=height, y=losses or x=volume, y=losses.
Response 10
We compared only volume change in felling and sawing stage. So, we used boxplot to show variations in actual felled tree volume, final bucked logs and timber losses in different stages. Actually we do not compare our data between three sampled sawmills or two forests. Also, we do not compare timber losses in different diameter classes or height/length classes. Boxplot graphically depicts groups of numerical data through their quartiles. We tried several times to represent our data in scatter plot and other forms but they cannot reflect our results well. Among them, boxplot showed clear range of timber losses in different stages of tree harvesting. So, we choose to use box plot.
In our study, we focus on timber volume losses in different harvesting stages. So, we used boxplots in this specific case study. It would be more informative to compare with DBH classes or height/length classes and use of scatter plot to show different data as you suggested.
Thank you for pointing it out. We made a mistake and put "barplots" instead of "boxplots". Now, we changed "barplots" to "boxplots" in figure 3 and figure 4. Now it reads as " Figure 3. Box plots showing variations in actual felled tree volume, final bucked logs and timber losses in tree felling stage." and " Figure 4. Box plots showing variations in actual felled log volume, final outturn volume and timber losses in logs sawing stage."
Comment 11
Paired T test at 5% level of significance revealed that there is significant timber> loss in logs sawing stage with present felling practice (p-value = 2.2e-16).
Is this test really needed? No one doubts the existence of saw mill losses. I would advise you to remove this sentence.
Response 11
Thank you for your suggestion. Yes, as it is obvious that timber losses exist in logs sawing or in tree felling stage. We did paired T test at 5 % level of significance to make argument strong and check scientifically whether there is significant volume of timber losses occurs losses in different stages of harvesting or not.
Comment 12
Figure 4. Bar plots showing variations in actual felled log volume, final outturn volume and timber losses in logs sawing stage.
To illustrate the variability across log diameters and companies, you could provide ascatter plot of losses on the y axis and stump volume on the x axis, using a different colour for the different saw mills and a different point form for the logs with stemrot. Alternatively similar scatter plots could be made with x=basal area, y=losses orx=Length of log, y=losses.
Response 12
As written in comment 9, we compared only volume change in sawing stage. So, we used boxplot to show variations in actual felled tree volume, final bucked logs and timber losses in sawing stage. Actually we do not compare our data between three sampled sawmills. Also, we do not compare timber losses in different basal area classes or log length classes. Boxplot graphically depicts groups of numerical data through their quartiles, So, boxplot showed clear range of timber losses in sawing stage.
It would be more informative to compare with basal area classes or log length classes and use of scatter plot to show different data as you suggested. In our study, we focus on timber volume losses in different harvesting stages. So, we used boxplots in this specific case study.
Comment 13
Training on RIL for felling and bucking losses reduction and enhanced ergonomiccondition for sawing losses reduction were other two highly recommended strategies (Figure 6).
There are different aspects of RIL as you mentioned in the introduction (1) reduction of damage to neighboring trees and regeneration losses (2) increased timber recovery. You insist on the second aspect, but the first aspect is probably the most relevant for longterm sustainable forest management.
Response 13
As you suggested, first point "reduction of damage to neighboring trees and regeneration losses" is most relevant for long-term sustainable forest management. This study focuses on quantification of timber volume losses and identifying its reduction strategies in different stages of harvesting. So that reduction of damage to neighboring trees and regeneration losses as well as RIL is out of scope of this study. Thank you for your valuable comment. Study on reduction of damage to neighboring trees and regeneration losses and RIL looks very interesting as suggested.
Comment 14
Coming back to this on my comment to section 4.3.(Page 13, 1st Para)
The rates are considerably higher compared to our findings. Plausible reasons forthe higher rate of loss could be: i) the studies considered total tree volumeincluding branches volume, whereas this study considered stem/timber volume only asthe total volume of tree i.e. clean bole volume of a tree, ii) the studies includedtransportation losses which was excluded in our study, and iii) the recovery ratecalculation in the studies considered the multiple species (i.e. average of recoveryrate of the timber volume of multiple species) whereas we studied only the economically most valuable species in Nepal.
Can you distinguish between coniferous and broad leaves? What about plantation forestry?
Response 14
Broader research focusing coniferous and broadleaves tree harvesting as well as plantation forestry is needed. In this research, we studied only in a single and one of the most economically valuable species in Nepal named "Sal (Shorea robusta)". So, timber losses in different types of forest and different origin of forest are out of our scope. Your suggestion in different type and origin of forest seems very interesting and need broader study.
Comment 15
Those residues were caused due to the saw dust, Bakal (unfit and chips), stem rot andknots.
Do you need to mention bark removal and side slabs?
Response 15
Thank you for the suggestion. We changed saw dust, Bakal (unfit and chips), stem rot and knots to saw dust, Bakal (unfit and chips), stem rot, knots, bark removal and side slabs as suggested. Now the line reads as " Those residues were caused due to the saw dust, Bakal (unfit and chips), stem rot, knots, bark removal and side slabs."
Comment 16
Table 10 Terai Sal forest, Nepal 30.81% (with rot) 29.72% (excluding rot)
This StudyAre these with/without rot loss rates consistent with the loss rates specified page 8
"Out of 167 sampled logs, 33 (19.8%) had stem rot. The percent of the stem volume> infected by the heart wood rot ranged widely among trees with stem rot (2.2% to> 48.1%). The loss averaged 1.54% across all logs sawed."
Response 16
Thank you for your suggestions. We have changed 21.59 % losses rate and 23.58% losses rate after deducting stem rot in table 9. But in table 10 it is correct as in table 8. Also we added "after deducting stem rot" and deleted "with rot" and "without rot" in table 10. We deleted "The loss averaged 1.54% across all logs sawed." in section 3.3 to reduce misleading as suggested.
Comment 17
Page 12
chair or table legs and tops which could use outfit sawn wood too.
Clarify the meaning of "outfit sawn wood".
Response 17
Thank you for your query. We mean to write use of sawn wood to the smallest size also. So we changed word "outfit sawn wood" to "smallest piece of sawn wood".
Comment 18
The use of conventional saw mills results in **severe** timber losses during>transportation and sawing (See 4.3)
Temper this sentence since losses are actually lower than in many countries.
Response 18
Thank you for your suggestions. The word "severe" is misleading. We have changed the sentence " The use of conventional saw mills results in severe timber losses during transportation and sawing" to the sentence "The use of conventional saw mills results in higher amount of timber losses during transportation and sawing in comparison of using portable saw mills in felling site".
Comment 19
Sal forests of Nepal are located in flat land which can easily accessed by mechanized harvesters.(page 15, 1st para)
Please mention the trade-off with soil damage potentially caused by mechanizedharvesters. Are harvester improving RIL mentioned in the introduction?
Response 19
RIL and soil damage potential are out of our scope. But, your suggestion in trade-off with soil damage potentially caused by mechanized harvesters and its effect in RIL seems interesting.
Comment 20
RIL demands substantial resources (financial, human and logistics), and sometimes is> also criticized as ‘Reduced Income Logging’ for the few years in the beginning (page 15).
RIL can be more demanding on the harvesting companies. But forest owners typically needto consider the whole growth cycle. Taking into account RIL's impacts on regeneration and neighboring trees. From an economic perspective, you could provide an estimation of the discounted future value of the losses due to CL. Such value increase could improve forest owner's financial incentive to perform RIL. RIL's improvements of future income are not assessed in this study, but you could make a better attempt at providing insights from the literature.
Response 20
Comparison of RIL and CL as well as economic perspective are out of our scope. But, your suggestion in estimation of the discounted future value of the losses due to CL seems interesting.
Comment 21
By using portable saw mills in the forest, villagers will be able to decide about where and how much to fell which will **restrict over logging**.(page 15, last para)
How can a tool prevent over harvesting? I think this sentence is incorrect. There are probably policies in place such as local or national forest codes that restrict over harvesting. If yes, these regulations should be mentioned here as well as potential enforcement issues.
Response 21
Thank you for your feedback. We completely agree that local or national forest codes or regulations could prevent over harvesting, if there is compliance and effective monitoring of harvesting process.
Now, we make changes in a given sentence to make it clearer as you suggested. Now the sentence reads as "By using portable saw mills in the forest, losses occurred during transportation and skidding can be minimized which can restrict over logging."
Comment 22
Ergonomics aimed at reducing the costs associated with equipment downtime, underutilization/overexploitation of forest resources, suboptimal processing capacity,> and other related problems which results in higher long-term benefits
This sentence doesn't make sense, separate it in two or remove it.
Response 22
Thank you for your suggestion. We removed the sentence as suggested.
Comment 23 (Conclusion Chapter)
a **considerable** amount of timber is lost while felling trees in Nepal. However, losses are lower than in many other world regions according to table 9.
Efforts paid in reducing logging and sawing losses can increase economic potentialthrough enhancement of timber recovery rate as well as the secondary use of thelogging and saw mill residues.
What is the evidence to support this statement?
Response 23
Our study showed that there is a significant timber volume loss in tree felling stage. So we used a word "considerable" amount of timber is lost while felling trees in Nepal.
We changed a sentence which reads as "Efforts paid in reducing logging and sawing losses can increase economic potential through enhancement of timber recovery rate." We removed the secondary use of logging and saw mill residues from the sentence which was more informative than our scope. Thank you for pointing out our over informative works.
Comment 24
The current study only studies CL. It would be interesting to compare CL and RIL in a
follow-up study.
Response 24
Yes, it seems interesting. Thank you for your suggestion.

Reviewer 3 Report
The manuscript deals with the identification and quantification the timber losses in different stages of tree conversion in the Terai region of Nepal, as well as the proposal of strategies to reduce timber losses in the tree harvesting process for Terai Sal (Shorea robusta) wood species. Some recommendations are also proposed as follows: 1) exploring and comparing appropriate harvesting equipment for felling, bucking and sawing taking all aspects (i.e. social, ecological and economical aspects) into consideration, 2) cost and benefit analysis of new equipment introduction, and 3) reduced impact logging covering wide forest management regimes in the Terai region of Nepal.
The manuscript is well-organized, clearly presented and discussed. It describes a lot of experimental work in the field and correlates the findings with other similar studies from different countries. The references are appropriate and related to the investigated topic.
Author Response
Comment
The manuscript deals with the identification and quantification the timber losses in different stages of tree conversion in the Terai region of Nepal, as well as the proposal of strategies to reduce timber losses in the tree harvesting process for Terai Sal (Shorea robusta) wood species. Some recommendations are also proposed as follows: 1) exploring and comparing appropriate harvesting equipment for felling, bucking and sawing taking all aspects (i.e. social, ecological and economical aspects) into consideration, 2) cost and benefit analysis of new equipment introduction, and 3) reduced impact logging covering wide forest management regimes in the Terai region of Nepal.
The manuscript is well-organized, clearly presented and discussed. It describes a lot of experimental work in the field and correlates the findings with other similar studies from different countries. The references are appropriate and related to the investigated topic.
Response 1
Thank you for such an encouraging review.